# Sociality and interaction envelope organize visual action representations

Leyla Tarhan [1]✉ & Talia Konkle [1]

Humans observe a wide range of actions in their surroundings. How is the visual cortex organized to process this diverse input? Using functional neuroimaging, we measured brain responses while participants viewed short videos of everyday actions, then probed the structure in these responses using voxel-wise encoding modeling. Responses are well fit by feature spaces that capture the body parts involved in an action and the action's targets (i.e. whether the action was directed at an object, another person, the actor, and space). Clustering analyses reveal five large-scale networks that summarize the voxel tuning: one related to social aspects of an action, and four related to the scale of the interaction envelope, ranging from fine-scale manipulations directed at objects, to large-scale whole-body movements directed at distant locations. We propose that these networks reveal the major representational joints in how actions are processed by visual regions of the brain.

[1] Department of Psychology, Harvard University, 33 Kirkland St., Cambridge, MA 02138, USA. ✉email: ltarhan@g.harvard.edu

We witness a multitude of actions in daily life: running, jumping, cooking, cleaning, writing, and painting, to name just a few. How does the brain make sense of this diverse input? The process begins with basic perception, forming lines and shapes into bodies in motion with identities and kinematic properties, and ultimately derives rich representations about emotional states, social interactions, and predictions about what will happen next[1–3]. Much previous work on the nature of action representation in the brain has focused on the end-stages of this process; for example, attempting to localize the more abstract and conceptual aspects of action representation[4–8]. However, recent research has begun to examine action-related processing from a more perceptual angle, asking how regions involved in high-level vision are organized to support action observation[3,9]. The present work follows this latter approach, leveraging an encoding analytical framework to characterize action responses in the visual system[10–12].

To employ this framework, we considered three theoretical questions and their methodological implications. The first relates to the nature of the domain in question: what do we mean by "actions" and how should we sample this domain representatively[13]? Many prior studies have used the tight link between actions and verbs as a guide, and have measured responses to written verbs or used verbs to guide the selection of experimental stimuli[5,14,15]. However, common verbs do not always have a clear corresponding visual action (e.g., "can"), and this approach may build in theoretical assumptions that actions that look very different but are described by the same verb share a neural representation (e.g., "pushing button" vs. "pushing a person"[16]). To approach action representation from a visual-perceptual level, we instead sampled our action stimuli based on common human experiences, using the American Time Use Survey as a guide[17] (see "Methods"). These actions span a wide range of activities, such as cooking, traveling, exercising, and recreating, that a large set of Americans reported engaging in on a daily basis.

A related consideration is how best to depict these actions in order to probe their underlying neural representations at a meaningful level of abstraction. To date, some researchers have used a small number of highly controlled action video stimuli[7,9,18–20], which help to isolate actions from their backgrounds, but constrain researchers' ability to discover joints in the broader domain of action perception. Others have taken a highly unconstrained approach, using rich, complex stimuli such as feature-length films[11,21,22]. This approach comes much closer to reflecting our daily perceptual experience with actions; however, too much complexity can make the resulting data challenging to wrangle into interpretable results. Thus, in the present work, we took an approach that sits in between these extremes: we selected a targeted and diverse subset of everyday actions, depicted using complex, heterogenous videos of a short duration.

The second major consideration concerns which brain regions to target. Prior work has revealed that watching other people's actions engages a broad network of regions with nodes in all major lobes of the brain, known as the "action observation network"[2,23–25]. However, the perceptual processing mechanisms that operate over objects, bodies, and motion likely also play a role in representing ecological actions. Therefore, it is critical to consider more widespread action responses across broad swathes of occipitotemporal and parietal cortices[9,16]. Given this, we developed a novel method to identify voxels that reliably differentiate among different actions[26], rather than constraining our analyses to specific regions of interest emphasized in the literature.

The final major consideration concerns the nature of the tuning in these regions. That is, what is it about an action that makes a given voxel respond more to that action than to others?

For example, actions differ in the "means" by which they are performed and "ends" or goals that they are accomplishing. Paralleling this division, regions of the lateral temporal cortex are tuned to body parts and postures[27–29], while inferior parietal sulcus contains information about actors' goals[30]. Others have argued that an action's sociality (whether or not an action is directed at a person) and transitivity (whether or not it is directed at an object) organize action processing in the lateral temporal cortex[3,9]. Drawing from this prior work, here we examined patterns of tuning to the different body parts that are involved in performing an action and what the action is directed at (its target; e.g., an object, another person, et cetera).

With these considerations in mind, the goal of this study was to understand and characterize what properties of actions best predict corresponding neural responses, allowing for the possibility that different regions of cortex are sensitive to different properties. To preview, we find that the effectors used to perform an action and what an action is targeted at successfully predict responses to action videos in much of the occipito-temporal and parietal cortices. Further, an analysis of the voxel-wise tunings to these features reveals a large-scale organization of five networks that span the ventral and dorsal visual streams. The tuning of these networks is related to actions' spatial scale of interaction ("interaction envelope") and relevance to agents ("sociality"). We argue that these networks reflect meaningful divisions in how actions—and, more broadly, visual inputs across domains—are processed by the brain.

## Results

**Stimuli spanning common actions**. To investigate neural responses to a wide range of human movements, 60 actions were selected based on what a large sample of Americans reported performing on a daily basis[31]. These actions span broad categories such as personal care, eating and drinking, socializing, and athletics (see https://osf.io/5qk8j/ for a figure summarizing these videos). Two short (2.5 s) videos were selected to depict each action, then divided into two sets. Each video depicts a sequence of movements (e.g., stirring a pot) rather than an isolated movement (e.g., grasping a spoon, taking one step), and thus may alternatively be thought of as depicting "activities" rather than "actions." However, "activity" can also connote something done over a much longer timescale and with greater variation in perceptual properties, such as "cooking dinner." Thus, for clarity we refer to the stimuli as "action" videos. Participants ($N = 13$) watched these action videos while undergoing functional magnetic resonance imaging (fMRI) in a condition-rich design that enabled us to extract whole-brain neural responses to each video (see Supplementary Methods).

**Voxel selection reveals reliable regions**. Before analyzing the structure in these neural responses, we first asked which regions of the brain reliably differentiate the actions at all, using reliability-based voxel selection[26] (Fig. 1). This selection method is well-suited to our voxel-wise modeling approach for two reasons. First, it retains voxels that show systematic differences in activation across the 60 actions, removing less reliable voxels and voxels that respond equally to all actions (see "Methods"). Second, this method requires voxels to show similar activation levels across the 60 actions in video set 1 and video set 2–thus, selected voxels necessarily have some tolerance to very low-level features given that the videos in each set differed in many respects (including the direction of movement, background, identity of the actor, et cetera).

The split-half reliability for each voxel is plotted in Fig. 1c (see Supplementary Fig. 2 for single-subject data). To select a cutoff

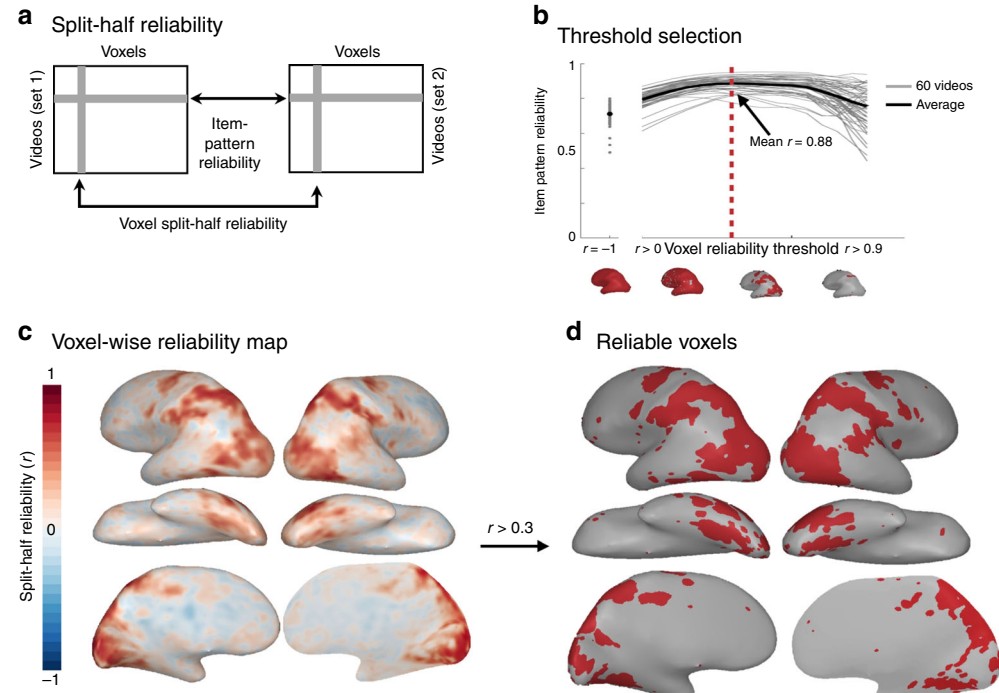

**Fig. 1 Reliability-based voxel selection. a** Schematic illustrating how voxel split-half reliability and item-pattern reliability were calculated from whole-brain response data. **b** Plot of average item-pattern reliability (split-half reliability of each item's multi-voxel pattern; y-axis) among voxels that survive a range of reliability cut-offs (x-axis). Brains along the x-axis display the voxels that survive the reliability cut-offs at $r = -1$, 0, 0.35, and 0.7. **c** Whole-brain map of split-half voxel reliability. **d** Reliable voxels ($r > 0.30$) selected based on the point where the curve plotted in (**b**) begins to plateau (see Tarhan and Konkle[26] for details). These results are based on group data. All analyses were conducted over reliable voxels, which were selected using the procedure outlined here. All brain figures were created by the authors.

for which voxels count as "reliable" voxels, we swept through a range of possible thresholds to find one that maximized both coverage and reliability[26] (see "Methods"). Through this procedure, a voxel-reliability cut-off of $r \geq 0.30$ was selected, yielding an average item-pattern reliability of $r = 0.88$ in the group data (Fig. 1b). This method revealed reliable activations along an extensive stretch of the ventral and parietal cortices, with coverage in lateral occipito-temporal cortex (OTC), ventral OTC, and the intra-parietal sulcus (IPS) (Fig. 1d). Reliability was relatively low in early visual areas, as expected from the cross-set reliability computation. All subsequent analyses were performed on this subset of reliable voxels.

**Operationalizing feature spaces with behavioral ratings.** Among the possible features that differentiate actions, we hypothesized that the body parts involved in an action and what an action is directed at are important dimensions that underlie at least some of the neural responses to actions. To measure the body parts feature space, human raters completed an online experiment in which they selected the body parts that were engaged by each action from among 20 possible effectors, such as legs, hands, eyes, torso, and individual fingers. The final feature values for each action were averaged over participants and are depicted for an example action in Fig. 2b. Given the natural covariance between different effectors, we used a principle components analysis (PCA) to reduce this feature space into seven principle components (PCs), which together account for 95% of the variance in the ratings (see Supplementary Fig. 1a). For example, body part PC1 distinguishes between actions that engage the legs (e.g., running) and those that engage the hands (e.g., painting).

Our second hypothesized feature space captures information about "action targets", i.e., what an action is directed at. To measure these features, raters answered questions about whether the action in each video was directed at an object, another person, the actor, the reachable space, and a distant location. Actions could have multiple targets. These ratings were averaged across participants and are shown for an example video in Fig. 2c. Following a PCA, all five PCs were needed to account for 95% of the variance in the ratings, where for example, the first PC distinguishes between actions that are directed at an object (e.g., shooting a basketball) and those that are directed at another person or the actor (e.g., shaking hands or running) (Supplementary Fig. 1b).

Together, the seven body-part principle component dimensions and five action-target principle component dimensions were combined into a 12-dimensional feature space that was used to model brain responses to the action videos. The dimensions from the body part and action target feature spaces were not well-correlated with each other (mean $r = 0.02$, range $= -0.40$ to 0.45; Supplementary Fig. 1c).

**Voxel-wise encoding models predict action responses.** The primary question of this study is whether these hypothesized feature spaces characterize neural responses to actions well, and if so where. To answer this question, we employed a voxel-wise encoding-model approach[10,11], which measures how well each voxel's tuning along these feature dimensions can predict its response to a new video. For example, a voxel that responds strongly to videos of sautéing and hammering but not to jumping will be best fit by high weights on hand, arm, and object-directed features, and low weights on the leg and actor-directed features. This model can then predict new responses; for example, the

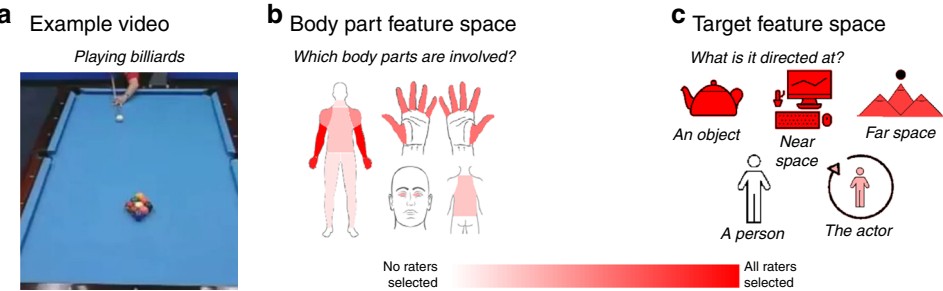

**Fig. 2 Feature spaces.** For each action video, the body parts and action targets involved in the action were estimated through online behavioral ratings experiments. **a** Keyframe from an example video depicting the action "playing billiards." This image was taken from the Human Motion Database (Creative Commons License CC BY 4.0, https://creativecommons.org/licenses/by/4.0/) and cropped for this experiment. **b** Subjects indicated which parts of the body were involved in the action video using a clickable body map. Color saturation indicates the number of participants who responded that each body part was involved in this example video. **c** Subjects rated the actions' targets by answering the following yes-or-no questions, aimed at one of the five possible targets: object: "Is this action directed at an object or set of objects?"; near space: "Are the surfaces and space within this actor's reach important for the action being performed?"; far space: "Is a location beyond the actor's reach important for the action being performed?"; another person: "Is this action directed at another person (not the actor)?"; the actor: Is this action directed at the actor themselves?". Color saturation indicates the number of participants who responded that each target was involved in this example driving video. Icons used to depict body part and target features were custom-made or based on images purchased from the Noun Project (Creative Commons License CC BY 3.0, https://creativecommons.org/licenses/by/3.0/), which were then colored and arranged by the authors.

same voxel should respond more to knitting than to running. The quality of the model's fit was assessed for each voxel based on how well it made these predictions for held-out action videos (measured as a high correlation between the actual and predicted responses to the held-out data; $r_{CV}$; see "Methods"). Figure 3 (inset) illustrates how well the neural responses in reliably responding voxels were predicted by fitting tuning weights to body part and action target features. Voxel responses were predicted well across much of the ventral and dorsal streams (median $r_{CV\text{-set1}} = 0.41$ and $r_{CV\text{-set2}} = 0.42$; max $r_{CV\text{-set1}} = 0.76$. $r_{CV\text{-set2}} = 0.71$; see Supplementary Fig. 4 for results in individual participants).

Critically, the key advantage of this encoding model framework is that we can examine not just how well the model predicts a voxel's response to a new action, but also why. For example, some voxels might be tuned to leg and foot involvement, while others might be tuned to hand and mouth involvement. To understand how these feature tunings are mapped across the cortex, we used k-means clustering to group voxels with similar feature tunings together (e.g., voxels assigned high weights on hand involvement and object targets but not leg involvement might be grouped together). Importantly, this method does not require that voxels are grouped into contiguous clusters, making it possible to discover both contiguous regions and networks of non-contiguous regions that have similar tuning functions. Further, this method does not presuppose any particular combinations of feature tunings ahead of time, allowing natural patterns to be revealed directly from the data (see Lashkari et al.[32] and Vul et al.[33] for related analysis approaches). Through this method, we found evidence for five networks (see "Methods", Supplementary Fig. 3a). Each network is shown with its corresponding tuning function in Fig. 3.

Network 1 (pink) is primarily right-lateralized, covering regions along the fusiform gyrus and extending between the occipital face area (OFA) and superior temporal sulcus (STS). This network is tuned to face features and not hands and is directed at people (others and the actor) but not objects. It is near STS regions typically engaged by social processing[34–36] while also extending inferiorly into ventral OTC along the fusiform gyrus. This network's tuning pattern highlights a possible large-scale neural division between social or body-centric actions that are directed at other people (shaking hands) or the actor themselves

(laughing), and those directed at objects and space. In fact, this division emerges early on: Network 1 separates out from the rest when voxels are grouped into just two networks (Supplementary Figs. 5, 6). This pattern suggests that the distinction between social or agent-focused actions and nonsocial actions is the predominant joint organizing action responses in these regions.

The remaining four networks are tuned to non-social aspects of action. The dorsal stream responses divided into two prominent networks. Network 2 (blue) contains voxels that stretch extensively along superior IPS, as well as a "satellite" node in lateral OTC. Inspecting the feature weights associated with this network indicates that these voxels respond most strongly to videos that involve finger and hand movements, and that are directed at objects in the near space (e.g., knitting). This finding resonates with a previously established tool network[3,9,24,37–40]. Network 3 (dark green) contains voxels that stretch extensively along inferior IPS, into the transverse occipital sulcus (TOS), with a satellite node in the parahippocampal cortex (PHC). Inspecting the feature weights associated with this network also reveals strong tuning to both objects and the near space, but with more arm (and not finger) involvement.

Network 4 (purple) is restricted to the ventral stream, including bilateral regions in the vicinity of extrastriate body area (EBA). Regions in this network are tuned to hands, arms, the torso, and near space.

Finally, Network 5 (light green) has nodes along PHC, TOS, and the medial surface of the cortex in both the parietal and retrosplenial regions. This final network effectively runs parallel to Network 3 (dark green): both networks anatomically resemble scene-preferring regions, such as the parahippocampal place area, occipital place area, and retrosplenial cortex[41–43]. However, compared with Network 3, Network 5 is tuned more strongly to actions directed at far space, the legs, and the whole body and less strongly to actions directed at objects. These differences in tuning echo prior work showing that at least one scene-preferring region—PPA—contains sub-regions for object and space processing[40].

Inspecting these last four non-social networks reveals a possible overarching organization, where each network has a preference for a different "interaction envelope", or the scale of space at which that agent-object interactions take place in the world. The four interaction envelopes move from small-scale,

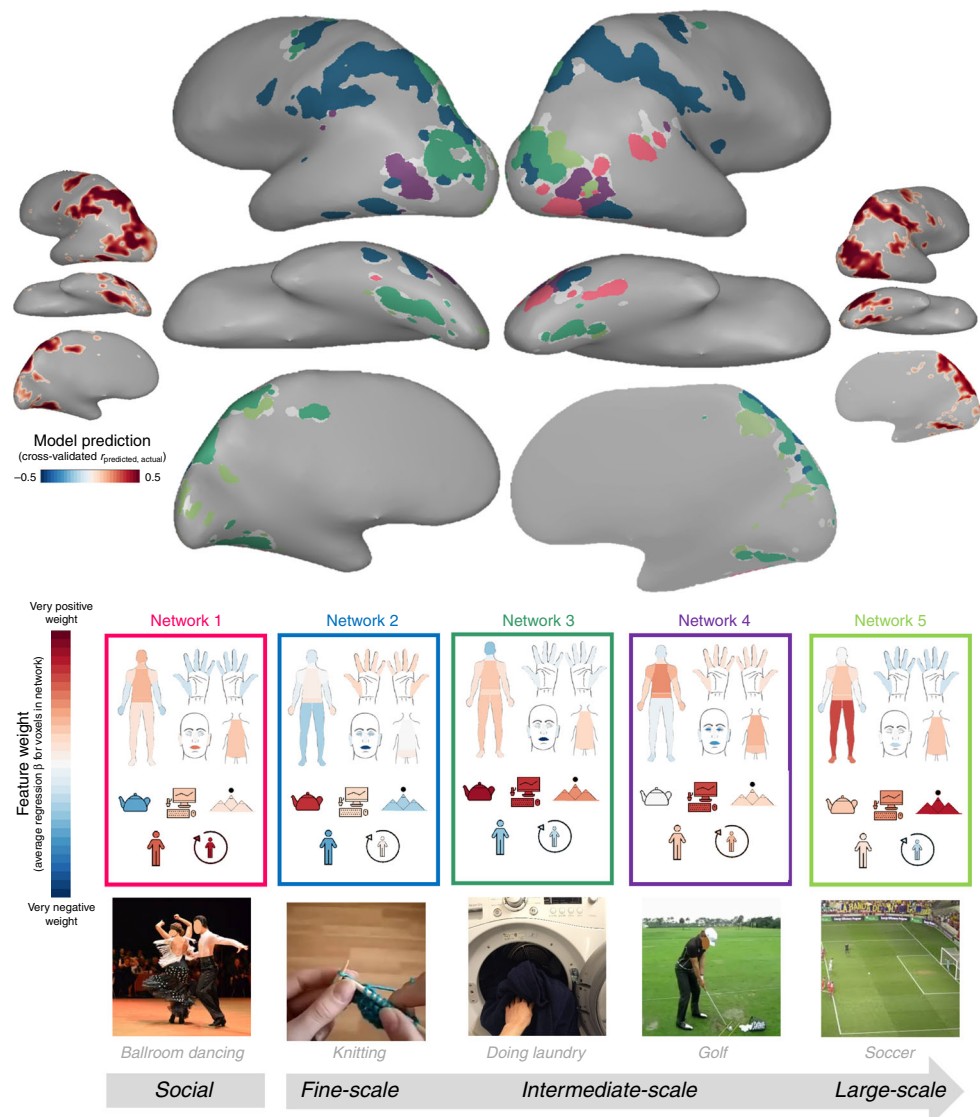

**Fig. 3 Large-scale feature tuning structure.** Data-driven clustering of the feature weights for body parts and target features. Inset brains display the prediction performance ($r_{CV}$, the correlation between predicted and actual responses in held-out data) for the combination of these features within reliable voxels. Larger brains display the five large-scale clusters on an example subject's brain. Heat maps display each cluster centroid's feature weight profile. Still images illustrate example actions that elicited high responses from each network. Images were taken from the Human Motion Database (golfing and knitting; Creative Commons License CC BY 4.0, https://creativecommons.org/licenses/by/4.0/), pxhere.com (ballroom dancing; Creative Commons License CC0 1.0, https://creativecommons.org/publicdomain/zero/1.0/), Michael Barera for Wikimedia Commons (soccer; Creative Commons License CC BY-SA 4.0, https://creativecommons.org/licenses/by-sa/4.0/deed.en), and author photographs (laundry) and then cropped. Results are displayed for video set 1 (see Supplementary Fig. 3 for video set 2 results). Only voxels that were well-fit by the model in both video sets (positive $r_{cv}$ that was significant after FDR correction for multiple comparisons) were analyzed. Voxels surviving this criterion with insufficient variance to be included in the clustering analysis are colored gray. All brain figures were created by the authors. Icons used to depict body part and target features were custom-made or based on images purchased from the Noun Project (Creative Commons License CC BY 3.0, https://creativecommons.org/licenses/by/3.0/), which were then colored and arranged by the authors.

precise movements involving hands and objects (knitting), to less precise movements also involving hands and objects (loading a washing machine), to intermediate-scale movements involving the upper body and near spaces (golfing), to large-scale movements that engage the whole body and require more space (playing soccer).

To examine the robustness of these findings in video set 1, we also conducted the same analysis on voxel tuning weights fit using video set 2. This five-network solution was quite consistent across the two video sets (cross-sets d-prime: 1.6; Supplementary Figs. 3b and c), pointing to the robustness of this network structure. In

general, we also found that the single-subject data echoed these patterns (Supplementary Fig. 4). However, note that not all subjects reflected the group data perfectly. In general, subjects with more extensive reliable coverage were a better reflection of the group, while those with fewer reliable voxels showed greater variability.

Taken together, these voxel-wise modeling and clustering analyses provide evidence for five networks that distinguish predominantly between social and non-social actions, then further divide the non-social actions into four large classes that vary in the scale of space at which they affect the world. These

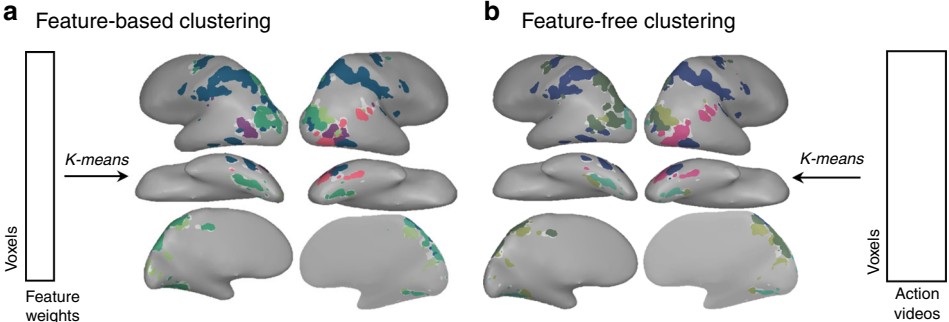

**Fig. 4 Generality of the large-scale structure.** We compared the results of clustering voxels based on the feature weights fit by the encoding models (**a**) and by the raw activation patterns across all 60 videos (**b**). The 5-network solutions over data from video set 1 are shown for both clustering analyses. Parts of cortex with similar colors had either similar feature weights (**a**) or similar overall response profiles (**b**). Each cluster's color was assigned algorithmically such that clusters with similar response profiles were similar in hue and were determined separately for (**a**) and (**b**). All brain figures were created by the authors.

data-driven analyses reveal that the relevance to agents and scale of an action in the world may be critical organizing factors for the perception and representation of actions in the brain.

**Auxiliary questions.** We next ask a series of more targeted questions to clarify the implications of these results and link them directly to related work.

First, how dependent is this network solution on the feature spaces? To arrive at the conclusion that there are five-major subnetworks underlying visual action perception, we relied on our hypothesized feature spaces, which characterize actions' body part involvement and targets. However, it is also possible that there is more systematic structure in the responses to these actions that is not captured by these feature spaces. In this case, the five networks we find may be only a partial reflection of the true subnetworks patterning this cortex.

To examine this possibility, we grouped voxels based on their profile of responses to all 60 videos in a stimulus set, rather than the profile of their 12-feature weights. Thus, the resulting clusters are driven entirely by the brain's responses to the videos themselves. The results of this analysis are displayed in Fig. 4. We again found evidence for five networks (cross-sets d-prime: 1.7) that recover a relatively similar structure to that observed based on the feature tuning (d-prime between feature-based and feature-free solutions: 1.6 for both video sets). The convergence between the results of the feature-free and feature-based analyses provides empirical evidence that the large-scale structure described above is not merely an artifact of the features we chose (see also Supplementary Fig. 7).

Second, what is the extent of lower-level feature tuning across this cortex? For example, videos vary in whether the actor is on the left or the right of the frame, and generally each video has a different spatial distribution of visual information across the frames. How well do these lower-level aspects of the videos' visual structure capture the structure of neural responses to actions?

Due to methodological considerations, motion features were not investigated in-depth in this study (see Supplementary Methods for further discussion). However, we did examine the role of low-level retinotopic image statistics using the Gist model[44]. We found that Gist features predicted brain responses well in early visual cortex, while the body part and target features provided better fits across occipitotemporal and parietal cortex (Fig. 5). Notably, the Gist features also fit moderately well throughout the dorsal and ventral streams, indicating that the action representations across this cortex retain information about the spatial layout of the action scene, even as higher-level features begin to emerge.

We also examined the possibility that the four networks tuned to different interaction envelopes may actually reflect tuning to the extent of motion in the videos (i.e., how much of a video's frame contains motion). We found that the spatial extent of motion in a video does track the size of the interaction envelope for small and intermediate interaction envelopes, but not the largest (Supplementary Fig. 8). Broadly, this line of inquiry raises further questions about what lower-level visual features predict the size of an action's interaction envelope, which will be fruitful ground for future research.

Third, are these neural responses simply reflecting what's visible in each video? The body parts and action targets that are involved in an action are undeniably related to what is visible in its video. However, we found that ratings of the features' visibility and involvement were not perfectly correlated ($r = 0.44$). Figure 6a shows the features that differed the most between ratings of their visibility and involvement. In some cases, the involved features were a subset of the visible features—for example, reading involves the actor's eyes, hands, and arms, and the book, but the reading video depicts the actor's whole upper body, a reachable desk surface, and a distant scene through a window. In other cases, involved body parts were off-screen—for example, vacuuming involved the arms, hands, and eyes, even though only the actor's legs and the vacuum were visible in the video.

When comparing how well these features predicted the brain, we found that the model based on the features involved in the actions out-performed the model based on the features that are visible in the videos in the posterior lateral occipital cortex, lateral temporal cortex in the vicinity of EBA, and the fusiform gyrus. In contrast, and somewhat surprisingly, the model based on the visible features predicted responses better in much of the IPS (Fig. 6b). Note that this analysis does not reveal how much variance is unique to each model; in fact, we found that a standard variance partitioning analysis[12] produced unstable results and was therefore ill-suited to our data. Instead, this analysis indicates that the visible bodies, people, objects, and backgrounds play an important role in how these actions are processed in some regions. However, the functional involvement of these effectors and targets is a better description of the neural responses throughout the ventral stream.

## Discussion

Perceiving the actions of others is an essential capacity of the human visual system. We found that much of the occipitotemporal and parietal cortices responded reliably during action observation, and these responses were well-predicted by models

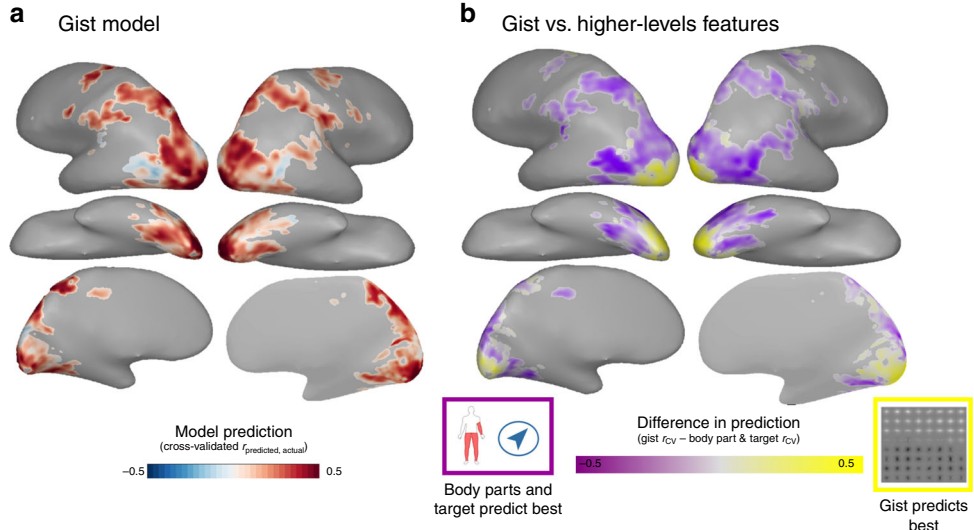

**Fig. 5 Incorporating low-level visual features. a** Voxel-wise encoding model prediction results for the gGist model in reliable voxels. Voxel color reflects the cross-validated correlation between predicted and actual response patterns to held-out items. **b** Two-way preference map comparing prediction performance for the gist features with performance for the body parts and action target features. Voxels are colored according to the model with the best cross-validated prediction performance: yellow for gist, and purple for the body parts and action target features. Color saturation reflects the strength of the voxel's preference ($r_{CV}$ for the gist model—$r_{CV}$ for the body parts and action target model). Icons used to depict body part and target features were custom-made or based on images purchased from the Noun Project (Creative Commons License CC BY 3.0, https://creativecommons.org/licenses/by/3.0/), which were then colored and arranged by the authors. All brain figures were created by the authors.

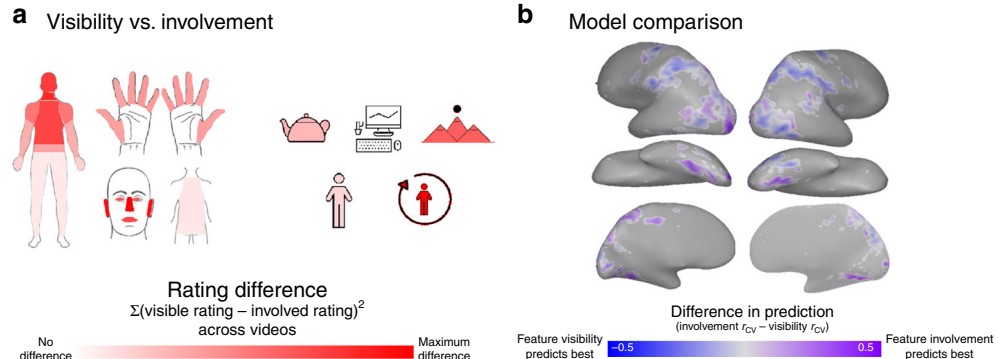

**Fig. 6 The role of feature visibility. a** Summary of feature-based differences between ratings of the features' visibility and involvement in the actions. Body parts and action targets that dissociate most strongly between ratings of visibility and involvement across our stimulus set are colored with the strongest saturation. Icons used to depict body part and target features were custom-made or based on images purchased from the Noun Project (Creative Commons License CC BY 3.0, https://creativecommons.org/licenses/by/3.0/), which were then colored and arranged by the authors. **b** Preference map comparing prediction performance for the models based on feature involvement and visibility. Voxels are colored according to the model with the best cross-validated prediction performance: purple for the features' involvement, and blue for their visibility. Color saturation reflects the strength of the preference ($r_{CV}$ for the involvement model—$r_{CV}$ for the visibility model). All brain figures were created by the authors.

based on the body parts engaged by the actions and their targets in the world. Examining the structure in voxels' tunings to these features revealed evidence for five action-processing sub-networks. The first of these networks was tuned to agent-directed actions, including actions directed at other people and the actor themselves. The remaining four networks varied in the scope of their "interaction envelope"—the spatial extent of interactions between agents, objects, and the environment. We propose that these five sub-networks reflect deeper joints within action processing in the visual system.

The five sub-networks found in our data converge with several known networks of the ventral and dorsal stream, including those that display preferences for bodies, faces, objects, and scenes[41,45–47]. For example, the agent-focused sub-network (Network 1) encompasses regions which are traditionally thought to be specialized for body parts, whole bodies, and faces[34,36,47–49], and is right-lateralized, consistent with prior results[50,51]. Our results also converge with previous findings that responses in OTC are organized by animacy and object size[52–54]. In general, big objects are processed in regions tuned to larger-scale actions, small objects are processing in regions tuned to smaller-scale actions, and animals are processed in regions tuned to agents. Additionally, the network related to large-scale interactions such as locomotion (Network 5) encompasses well-known scene-selective regions near occipital place area, retrosplenial cortex, and along the parahippocampal place area. Thus, our results dovetail well with existing characterizations of this cortical territory. However, note these regions are typically identified by contrasting responses to isolated bodies, objects, and scenes, all of which are present in our videos. Therefore, our results likely converge with prior

findings not simply because of the mere presence of these visual categories but because scenes and bodies are more relevant to some actions than others.

In some cases, our results diverge from prior work. For example, the sub-network related to large-scale actions and far spaces (Network 5), is primarily right-lateralized in both video sets. However, most research on scene networks shows relatively clear bilateral networks[41–43]. Relatedly, the sub-networks tuned to object targets within a more focal interaction envelope (Networks 2 and 3) are bilateral in this data set, which is consistent with some tool-network studies[21,28,55], but not others[37,38,56].

Perhaps the most surprising result was that ventral stream responses were predicted best by whether body parts and action targets were involved in the action, while dorsal stream responses were predicted best by their visibility in the video. While it is clear that viewing actions drives both the ventral and dorsal visual streams, it is also clear that more work is needed to understand how the streams relate to one another during action observation. One potential avenue for clarifying this relationship is to characterize the relative contributions of what is visible and what participants fixate on. It is possible that participants fixate on the features that are more functionally relevant to an action. If this is true, fixated features may be prioritized in ventral stream, while the dorsal stream may represent the full action scene to guide subsequent fixations. Future eye-tracking studies are needed to test these hypotheses.

In addition to converging with prior work on other categories of visual input, our methods build on the use of encoding models to predict responses to rich videos[11,22]. However, our approaches differ in the granularity at which we predict voxel responses and subsequently infer voxel tuning properties. In Huth et al.[11], a voxel's response could be fit by putting weights on over 1000 predictors, including verbs like "cooking", "talking", and "crawl", as well as nouns like "tortoise" and "vascular plant." It is possible to map some of these specific features onto our more-general ones—for example, a voxel in their data set with high weights on "knitting" and "writing" might be fit in our data set by high weights on the hand, fingers, and object-target features. However, one potential advantage of characterizing the brain's responses at the level of body parts and targets is that the features are more generative: it is simple to map any new action into this low-dimensional feature space, and this level of representation may therefore be more appropriate for characterizing the response tuning of mid-to-high-level visual cortex.

How do our findings relate to other work on the features that organize action representations? Our main finding—that the most predominant division in neural responses was between actions focused on agents and those focused on objects—is consistent with Wurm et al.'s[9] proposal that sociality and transitivity organize action representations. However, we did not find evidence for their anatomical proposal that these features are reflected in a ventral-dorsal organization across the lateral OTC (Supplementary Fig. 9). Rather, we found that the dorsal stream and parahippocampal gyrus are more sensitive to object-directed actions, while most of lateral OTC and the fusiform gyrus are sensitive to whether an action is directed at the actor or another person. More generally, our data-driven finding of a neural joint between agent-focused and non-agent-focused actions dovetails nicely with the broader argument that humans recruit fundamentally different cognitive architectures when processing agents (e.g., inferring beliefs and goals or detecting interactions between agents) and interactions that are less focused on the agents (e.g., reaching, grasping, and navigation)[19,57–59].

We also found that regions tuned to non-agent-focused actions branched into four networks. We propose that these highlight different interaction envelopes, or the scales at which actions affect the world. These range from fine-motor, hand-focused actions like knitting, to coarser movements like doing laundry, to intermediate actions involving the upper body and near spaces like golfing, to large-scale actions that move the whole body within a larger space, like playing soccer. To our knowledge, the term "interaction envelope" was first introduced in the visual cognitive neuroscience literature to highlight the difference between objects typically used with either one hand or two[60]. Here we have adopted this term and expanded its scope.

While the scale of the interaction envelope is a fairly intuitive continuum along which to organize actions, it is also a relatively novel theoretical proposal, contrasting with the more linguistic properties emphasized in the literature, such as transitivity and communicativeness[3,6,39]. And while action processing is often studied separately from object and scene processing, the concept of the "interaction envelope" requires integrating object, agent, and scene properties. For example, most tool-use actions are also hand-based and happen in near-space, while fitness actions tend to engage the whole body within a larger spatial envelope. From this perspective, perhaps actions are a chassis that connects the visual processing of objects, bodies, and space, rather than reflecting a separate domain of visual input.

## Methods

**Experimental model and subject details**. Thirteen healthy, right-handed subjects (five males, age: 21–39 years) with normal or corrected-to-normal vision were recruited through the Department of Psychology at Harvard University and participated in a 2-h neuroimaging experiment. In addition, 802 participants completed behavioral rating experiments conducted online. All subjects gave informed consent according to procedures approved by the Harvard University Internal Review Board.

**Stimulus set**. One hundred twenty 2.5-s videos of 60 everyday actions were collected from YouTube, Vine, the Human Movement Database[61], and the University of Central Florida's Action Recognition Data Set[62]. These were divided into two sets of 60 videos each, so that each set contained one exemplar depicting each of the 60 actions. All videos were cropped to a 512-by-512 px frame centered on the action, with no visible logos or borders, and stripped of sound. The 60 actions were selected based on the American Time Use Survey corpus[31], which records the activities that Americans perform on a regular basis, across a range of general categories (e.g., household chores, fitness, and work tasks). These categories were used to ensure that we sampled a wide range of everyday visual experience but were not used in the analyses. Key frames from both sets of videos are available for download from the Open Science Framework.

**fMRI data collection**. Participants viewed 120 videos of everyday actions in a condition-rich design while undergoing functional neuroimaging (Supplementary Methods). To ensure that they remained alert throughout the experiment, participants pressed a button whenever a red frame appeared around a video.

**Action feature ratings**. To collect action feature ratings, four behavioral experiments were conducted on Amazon Mechanical Turk. In all experiments, 9–12 raters viewed each video, named the action depicted, and answered experiment-specific follow-up questions. In Experiment 1 ($N = 182$), raters indicated the body parts that were engaged by the action in the video by selecting them from a clickable map of the human body: when a body part was selected, it was highlighted in red (Fig. 2b). There were 20 possible body parts: eyes, nose, mouth, ears, head, neck, shoulders, torso, back, arms, hands, individual fingers, waist, butt, legs, and feet. The lateralization of body parts was not analyzed: e.g., a rating was recorded for "hand" if raters selected the right hand, left hand, or both hands. In Experiment 2 ($N = 240$), raters indicated the body parts that were visible at any point in the video using the same method. In Experiment 3 ($N = 180$), raters indicated what each action was directed at by answering five yes-or-no questions: "is this action directed at an object or set of objects?"; "is this action directed at another person (not the actor)?"; is this action directed at the actor themselves?"; "are the surfaces and space within the actor's reach important for the action being performed?"; "is a location beyond the actor's reach important for the action being performed?" (Fig. 2c). In Experiment 4 ($N = 200$), raters responded to similar questions about the targets that were visible in the videos (e.g., "is an object or set of objects visible in the video?").

**Gist features**. For comparison with the body parts and action target features, the "gist" model[44] was included in the analysis as a measure of low-level visual

variability between videos. First, each video frame was divided into an 8 × 8 grid. At each grid location we quantified the power at 4 different spatial frequencies and scales (12 orientations at the finest scale, 8 at the intermediate scale, and 6 at the coarsest scale), yielding a 1,920-dimensional feature vector for each frame. These gist features were extracted for each video frame and then submitted to a PCA to obtain features that describe the global shape of the scene. Following Oliva and Torralba[44], the first 20 PCs were then averaged across frames in a given video, resulting in a 120-by-20 gist feature matrix. Together, these 20 PCs accounted for nearly 100% of the variance in the videos.

**fMRI reliability and voxel selection**. Split-half reliability was calculated for each voxel by correlating the betas extracted from odd and even runs of the main task (Fig. 1a). This was done in two ways. Reliability was calculated across sets by correlating odd and even betas from glms calculated over the two video sets. Across-sets reliable voxels did not extensively cover early visual cortex, as this scheme requires responses to generalize over two different exemplars of the same action. Reliability was also calculated within sets by correlating odd and even betas separately for each set, then averaging the resulting $r$-maps. Within-set reliable voxels had better coverage of early visual cortex and were only used to compare the gist model to the body-part-action-target model (Fig. 5b). For both types of reliability, we selected a reliability-based cutoff using a procedure from Tarhan and Konkle[26], which strikes a balance between selecting a relatively small set of voxels with the highest reliability and selecting a larger number of voxels but with lower reliability. First, we plotted the reliability of each video's multi-voxel response pattern ("item-pattern reliability") across a range of candidate cutoffs. Then, we selected the cutoff based on where the multi-voxel pattern reliability begins to plateau for all videos. This method takes advantage of the fact that after a certain point, using a stricter cutoff restricts coverage without significantly increasing the data's reliability. Using this approach, we determined that any voxel with an average reliability of 0.3 or higher was a reasonable cutoff for inclusion in the feature modeling analysis because it maximized reliability without sacrificing too much coverage (Fig. 1b, c). This cutoff held in both group and single-subject data; however, only voxels that were reliable at the group level were analyzed.

**Voxel-wise encoding modeling**. We used an encoding-model approach[10,11] to model each voxel's response magnitude for each action video as a weighted sum of the elements in the video's feature vector (e.g., individual body parts) using L2 ("ridge") regularized regression. Models were fit separately for the two video sets. The regularization coefficient ($\lambda$) in each voxel was selected from 100 possible values (ranging from 0 to the maximum value that produced a non-null model for that voxel). The final $\lambda$ was the value that minimized the mean-squared error of the fit in a 10-fold cross-validation procedure, using the data from the other video set. To ensure that our models were not over-fit, we estimated their ability to predict out of sample using a cross-sets cross-validation procedure. This was done by training the model in every voxel, using the data from all 60 videos in one video set. We then calculated the predicted response magnitude for the second video set (beta weights from the training model * feature vector for the held-out video set). Finally, the predicted and actual data for the held-out actions were correlated to produce a single cross-validated $r$-value ($r_{CV}$) for each voxel. All models were fit using responses from the group data (see Supplementary Fig. 4 for single-subject results).

**Data-driven neural clustering**. We used k-means clustering to group voxels based on their feature weight profiles. This analysis groups voxels based on the similarity of the weights that the voxel-wise encoding model assigned to the 12 body-part and action-target features. These feature weights were measured by fitting the model on the complete dataset for each video set, and the clustering analysis was conducted separately in both Set 1 and Set 2, enabling a internal replication of the main result. In order to facilitate a comparison between the sets, we ensured these clustering analyses were conducted over the same set of voxels—thus, we only included voxels that were reliable across both video sets and were predicted well ($r_{CV} > 0$, with $q < 0.01$ after FDR-correction) by both the models. Next, we used MATLAB's implementation of the k-means algorithm with the correlation distance metric to cluster voxels by feature weight profile similarity (10 replicates, 500 max iterations). The correlation metric was chosen in order to group voxels together that have similar relative weightings across the features (e.g., higher for leg involvement and lower for near-space targets). For any clustering solution, the cluster centroid weight profiles reflect the average normalized profile for all voxels included in the cluster.

To determine the number of clusters to group the voxels into ($k$), we iteratively performed k-means on the data from video set 1, varying the possible $k$ value from 2 to 20 (Supplementary Fig. 3). When choosing the final $k$, we considered both silhouette distance (how close each voxel is in the 12-dimensional feature-space to other voxels in its cluster, relative to other nearby clusters) and cluster center similarity (how similar the cluster centers are to one another on average). The logic for the latter measure is that if two clusters have very similar centroids, there is very little we can do to interpret their differences; thus, by considering solutions with lower cluster center similarity we are better equipped to interpret divisions within this feature space. In addition, we visualized the solutions at $k = 2, 3,$ and 4

(Supplementary Fig. 5), to provide insight into the hierarchical structure of these networks.

To visualize the final clustering solution, we created cortical maps in which all voxels assigned to the same cluster were colored the same. For visualization purposes, we chose these colors algorithmically such that clusters with similar response profiles were similar in hue. To do so, we submitted the clusters' feature weight profiles to a multi-dimensional scaling algorithm using the correlation distance metric, placing similar cluster centroids nearby in a three-dimensional space. The 3D coordinates of each point were re-scaled to fall in the range [0 1], and then used as the Red-Green-Blue color channels for the cluster color. We then plotted each cluster's centroid to determine how to interpret the groupings. This was done by multiplying the centroid's feature weight profile over the 12-feature PCs by the factor loading matrix from the PCA, effectively projecting the centroid back into the original feature space. The resulting weights over the 25 original features were visualized using custom icons, where each feature's color reflected the weighting of its associated PC feature in the centroid.

As a test of robustness, we calculated the sensitivity index (d′) between the solutions for video sets 1 and 2. To do so, we created a voxels x voxels matrix for both video sets, with values equal to 1 if the voxels were assigned to the same cluster and 0 if they were assigned to different clusters. Hit rate was calculated as the percent of voxel-voxel pairs that were assigned to the same cluster in set 1 that were also assigned to the same cluster in set 2. False alarm rate was calculated as the percent of voxel-voxel pairs that were not assigned to the same cluster in set 1 but were assigned to the same cluster in set 2. Sensitivity (d′) was subsequently calculated as z(Hit)-z(FA). D′ was also calculated between the shuffled data for sets 1 and 2, producing a baseline value close to zero (mean across $k$-values from 2 to 20 = −0.002, sd = 0.003). Finally, analogous cluster centroids were correlated between sets (Supplementary Figs. 3, 7).

To compare how the voxels cluster independently of their feature tunings, we followed a similar procedure to group voxels by their response profiles. Here, instead of using feature weights, we performed the clustering over voxels' response magnitudes to each action video. All other procedures were the same as those used for feature-based clustering.

**Preference mapping analyses**. To compare performance based on the low-level gist model to the higher-level body part and action target models across visual cortex, we calculated a two-way preference map. First, both the gist model and the body-part-action-target model were used to predict responses in within-sets reliable voxels. Within-sets reliable voxels were used to increase our coverage in the early visual cortex, where we anticipated the gist model would do best. Then, in each voxel, we found the model with the maximum $r_{CV}$ and colored it so that the hue corresponds to the winning model and the saturation corresponds to the size of the difference between that value and the alternative model's performance (e.g., a voxel where the gist model did markedly better than the body-part-action-target model is colored intensely yellow; Fig. 5b). The gist and body-parts-action-target features spaces were not well-correlated—the representational dissimilarity matrix correlation between them was $r = 0.01$ for video set 1 $r = 0.05$ for video set 2.

To compare models based on the functional involvement and visibility of body parts and action targets, we calculated a two-way preference map. As described above, each voxel was colored according to the model that predicted its responses the best (purple: feature involvement, blue: feature visibility; Fig. 6b) and its saturation reflected the strength of the preference.

**Reporting summary**. Further information on research design is available in the Nature Research Reporting Summary linked to this article.

## Data availability
Example frames for all stimuli, all feature ratings, and pre-processed group and single-subject fMRI data are available at the Open Science Framework repository for this project (https://osf.io/uvbg7/).

## Code availability
Code for the main analyses is available at the Open Science Framework repository for this project (https://osf.io/uvbg7/).

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

## Acknowledgements

Funding for this project was provided by NIH grant S10OD020039 to Harvard University Center for Brain Science, NSF grant DGE1144152 to L.T., and the Star Family Challenge Grant to T.K.

## Author contributions

L.T. and T.K. designed research; L.T. performed research; L.T. and T.K. contributed analytic tools; L.T. and T.K. analyzed data; L.T. and T.K. wrote the paper.

## Competing interests

The authors declare no competing interests.
