## [Peer Review File · Nature Communications]

Reviewers' Comments:

Reviewer #1:

Remarks to the Author:

Konkle and Tarhan examine the representations of actions across human cortex using fMRI and voxel-wise encoding models. The authors find that voxel activity is well predicted by a feature space describing the effectors and targets of actions, and the primary dimensions of this feature space are the sociality of an action and the size of the "interaction envelope". Overall I think this study addresses an important topic and is very well designed. I believe there are a few additional analyses - related to the clustering analysis and comparison of different feature spaces - that are critical to support their conclusions and several points in the paper that should be clarified. Specific comments below.

Major comments:

Voxel selection - Please provide more detail about how the threshold for voxel selection was chosen. In page 5, what does it mean to "maximize coverage and reliability"? I understand this is covered in a bit more depth in your referenced paper, but 1-2 sentences on how the 0.3 value was chosen (here or in the methods) seems important to evaluate the current results.

I think a better way to look at the hierarchical structure of the networks is with hierarchical clustering (rather than selecting different values of k in k -means). This is important to verify the central claim that social/non-social is the predominant organizing dimension of the data. This analysis can also reveal potential hierarchical structure in the remaining non-social networks. If a similar number/structure of clusters emerge that would also provide strong converging evidence for the paper's main claim.

When comparing how well different feature spaces explain your data (body-part-action-target vs. gist vs. visibility), it is important to account for the shared variance across these different feature spaces. I assume the body-part-action-target and gist models are uncorrelated (but the authors should report this) across the videos in the dataset. The effector/target and visibility models are fairly correlated though. In this case, I think it is important to perform commonality analysis to look at the shared and unique variance of these two models across cortex. Like the authors, I am similarly puzzled by the fact that the visibility model explains dorsal cortex results while the effector/target explains ventral pathway. This effect may be an artifact, though of not accounting for the shared variance across these feature spaces.

Minor comments:

There was a nice description of the term interaction envelope in the discussion, but the term seems a bit confusing to me. I think it would help to clarify that these refer to agent-object interactions rather than social agent-agent interactions.

Pg 4 Results paragraph 1 - The action categories were not used for selection or analysis purposes at all, correct? If so, please clarify in text (I see that the figure says they are for illustrative purposes).

Pg 8 - related to the above it is not clear in the main text how rcv is calculated across the two sets of videos. From reading the methods, I believe that rcv is calculated separately for each half of the data and then only voxels that are found to have $rcv > 0$ in both analyses are selected. Please clarify.

Pg 6 - For both action and effector PCA analyses you describe the first PC in detail. Were the other PCs similarly interpretable? I see the schematics in Figure S1, but it is a bit hard to interpret these alone.

Figure 4- what is r_{cv} ? Is it the same thing as the model prediction correlation presented in the top left? Please clarify in figure legend and text.

Pg 9 - Network 1 appears to be tuned only to mouths (not other face features) based on the coloring in Figure 4. Is this correct and what are the implications of this?

Reviewer #2:

Remarks to the Author:

This study by Tarhan & Konkle looked at the functional organization of visual actions across cerebral cortex in humans. They use voxel-wise encoding models to predict brain activity in response to held-out videos of humans performing everyday actions, and show that these models are strikingly similar across two sets of videos. In addition, they cluster the data in order to interpret the networks of areas that represent this information.

Overall, this study is quite thorough. The methods show clearly that the results are replicable. The results are interesting. However, the specific methods are suboptimal in several respects. Given that voxel-wise modeling is relatively rarely used in fMRI it is important that the methods used in any voxel-wise modeling report should be beyond reproach. I think it is likely that these suggestions will not change the overall results of the paper, but they will make the paper stronger.

Major Issues:

This is a vision science paper, and it has been standard practice in the vision community for many years to present all of the data from each individual subject. Showing data from individual subjects avoids confounds due to cortical normalization and it increases confidence in the results. This paper claims to be using the same voxel-wise encoding model pipelines as Mitchell et al. (2008) and Huth et al. (2012), so showing data from individual subjects should not be a problem. Given the amount of data collected on each subject, it should be possible to show the effect in each subject, and if not, the authors should report how many of the individual subjects do show the effect.

There are some major problems with how PCA is used in this study. Typically, during voxel-wise modeling PCA is used as a dimensionality reduction technique during data interpretation, in situations with a very large number of variables/features. However, in this paper PCA is used as part of the model fitting process. However, it is not necessary to use PCA to fit the body part and target feature spaces used here because they have only 20 and 5 features, respectively. All fitting and interpretation involving those features should be done in the native space, which were explicitly labeled by participants.

Because the Gist model has many parameters PCA might offer a way to reduce computation time during model fitting. However, if it is feasible to fit those models without PCA then this would be preferable. If PCA is needed later in order to interpret the feature tuning of voxels fit well by that model, then PCA should be run on the fit model weights, not the feature space.

Finally, if the authors find it impossible to fit the Gist model without using PCA, then they should justify the number of features used, as they did with the other feature spaces (e.g. 95% variance explained). In the current version of the manuscript, it is unclear whether the Gist model is fitting worse than the higher-level action feature model in some brain areas merely because the PC features do not capture enough of the variance in the specific features that are encoded in those areas. Note that this suggestion may reduce the number of voxels that prefer the higher-level model, but it is impossible to predict by how much these results might change.

The authors claim that they apply PCA to the stimulus features during fitting in order to remove correlations between variables. However, this is completely unnecessary because they use ridge regression for model fitting. Ridge regression deals with the problem of correlated variables directly. In fact, because ridge regression uses regularization and PCA does not, in the presence of noise the ridge

procedure is likely to do a better job than PCA. For this reason the PCA step should be removed from the modeling procedure. This will also make it easier to interpret the fit model weights in terms of the feature spaces of interest.

I expect that these recommended changes in the use of PCA will not impact the conclusions of they study, but there is a chance that they will increase model performance for all models. In any case, these analyses should be corrected in order to conform to best statistical practices.

When using regularized regression to fit models to noisy data, the regularization parameter often has the largest effect on the outcome. Therefore, it is critically important to ensure that the data set used for regularization and model fitting is completely separate from the data set used for evaluation of the fit models. However, as far as I can tell the current manuscript does not enforce this critical separation. Instead, the regularization coefficient through k-fold cross validation on the training data (which is fine), but then the model is evaluated based on the same data that was used to select the regularization coefficient (for one set of stimuli). The authors of the paper already have two sets of data, so it would be incredibly straightforward to fit the model on set 1 of the stimuli and then to use the second set of stimuli to evaluate the model.

In revision the authors should also provide a more convincing test of model performance. Separating fit and test data sets in this way has proven to be a much more robust test of a model than the leave one out cross validation (LOOCV) procedure that the authors used in the current submission. Again, the correct way to deal with this issue is to fit the data on set 1 of the stimuli and then predict responses to set 2 of the stimuli. Thus, separating sets 1 and 2 of the stimuli for model fitting and model testing would solve both of these problems simultaneously.

Finally, I should note that the threshold used for "well-fit" voxels appears to be rather arbitrary. It is currently defined as those with $rCV > 0$, but statistical significance should be determined empirically. The most straightforward way to do this would be to get the p-values associated with each voxel's rCV value. These values should then be FDR-corrected, and the voxels with values remaining above threshold should be further analyzed. This change should clean up the cluster analyses, and it is likely that fewer voxels will be colored gray in Figure 4 after this threshold is revised.

Minor Issues:

All figures with inflated brains should show both left and right hemispheres, as well as medial views. This is important not only for completeness, but also because the

paper makes reference to results in medial areas (such as retrosplenial cortex), which are not actually visible in any of the main figures of the paper.

It is unclear where the model weights for clustering come from, particularly because models are fit 60 times during LOOCV. I assume that these are the average model weights across all of these folds, but that is not stated explicitly in the paper. This information should be included in the methods.

From the methods, it seems that the body part features do not differentiate the left and the right sides of the body. If this is true then it should be added to the methods for clarity.

Response to Reviewers

Reviewer #1

Konkle and Tarhan examine the representations of actions across human cortex using fMRI and voxel-wise encoding models. The authors find that voxel activity is well predicted by a feature space describing the effectors and targets of actions, and the primary dimensions of this feature space are the sociality of an action and the size of the “interaction envelope”. Overall I think this study addresses an important topic and is very well designed. I believe there are a few additional analyses - related to the clustering analysis and comparison of different feature spaces - that are critical to support their conclusions and several points in the paper that should be clarified. Specific comments below.

Thank you for this positive feedback!

Major comments:

Voxel selection - Please provide more detail about how the threshold for voxel selection was chosen. In page 5, what does it mean to “maximize coverage and reliability”? I understand this is covered in a bit more depth in your referenced paper, but 1-2 sentences on how the 0.3 value was chosen (here or in the methods) seems important to evaluate the current results.

We have added to our explanation of this method in the Methods (pg. 18-19).

I think a better way to look at the hierarchical structure of the networks is with hierarchical clustering (rather than selecting different values of k in k -means). This is important to verify the central claim that social/non-social is the predominant organizing dimension of the data. This analysis can also reveal potential hierarchical structure in the remaining non-social networks. If a similar number/structure of clusters emerge that would also provide strong converging evidence for the paper’s main claim.

Following this suggestion, we performed a hierarchical clustering analysis. When voxels were grouped into two clusters, we found a very similar pattern to that observed using \$k\$ -means clustering—this result lends new empirical support to our speculation that the social/non-social division is the predominant joint in the five-network organization that we found. Interestingly, we found it was difficult to interpret subsequent divisions found by the hierarchical clustering analysis, because the next clusters formed were very small (e.g., 30 voxels or fewer), and did not provide deeper insight into the non-social networks. We have added this analysis to the Results and Supplement (pg. 9, 30-31, **Figure S6**).

When comparing how well different feature spaces explain your data (body-part-action-target vs. gist vs. visibility), it is important to account for the shared variance across these different feature spaces. I assume the body-part-action-target and gist models are uncorrelated (but the authors should report this) across the videos in the dataset.

In the revised manuscript we now report the relationship between the action-target-bodypart model and the gist RDM in the Results section (pg. 11). As you guessed, these were not well-correlated (\$r = 0.01\$ for set 1 and \$r = 0.05\$ for set 2).

The effector/target and visibility models are fairly correlated though. In this case, I think it is important to perform commonality analysis to look at the shared and unique variance of these two models across cortex. Like the authors, I am similarly puzzled by the fact that the visibility model explains dorsal cortex results while the effector/target explains ventral pathway. This effect may be an artifact, though of not accounting for the shared variance across these feature spaces.

We agree, and in fact we had previously conducted a Variance Partitioning Analysis to compare these models using a procedure outlined in Lescroart et al. (2015). However, this method yielded a large number of voxels that showed *worse* performance when fit with both models than either model on its own, leading to negative variance estimates! This phenomenon may be linked “variance suppression” effects. Thus, the standard variance partitioning method did not yield easily interpretable insights into this particular puzzle of the data. A summary figure of this analysis is below for your reference, and the procedure we followed to compute these numbers is detailed in the caption. We did not include this analysis in the revised manuscript, as we found it inconclusive, and more confusing than helpful. However, we think the results are still valid (if puzzling): when one model predicts a voxel better than the other, this is likely due to that model’s unique variance. However, in this dataset, it is not straightforward to estimate the exact amount of unique variance accounted for by each model, given how correlated they are.

Figure 1. Results of a Variance Partitioning Analysis to estimate the unique variance accounted for by each model. Here, the results are averaged over all voxels in four large sectors of the visual cortex and are displayed for each of the four feature spaces (involved and visible body parts; and involved and visible action targets). To compute the unique variance accounted for by each model in each voxel, we first used each model to predict neural responses separately, without any regularization. This produced one estimate of prediction performance (r_{CV}) for each

model, in every voxel, which we then squared (r_{cv}^2). Next, we did the same thing using a combination of the two feature spaces (concatenated together) to predict neural responses. Then, we estimated the unique variance that each model accounted for by subtracting the r_{cv}^2 values for the single model from the r_{cv}^2 values for the combination model. Finally, we estimated the variance shared by the models as: r_{cv}^2 for involvement + r_{cv}^2 for visibility - r_{cv}^2 for the combination. In theory, the combination of the feature spaces should predict the brain as well as or better than either model on its own; therefore, all estimates of shared and unique variance should be positive. However, we observed a large number of voxels with negative estimates. In these cases, the combination of the feature spaces was *worse* than either model on its own.

Minor comments:

There was a nice description of the term interaction envelope in the discussion, but the term seems a bit confusing to me. I think it would help to clarify that these refer to agent-object interactions rather than social agent-agent interactions.

Thank you for making this point. We have made several changes throughout the manuscript clarifying this distinction (e.g., significance statement, results, Discussion).

Pg 4 Results paragraph 1 - The action categories were not used for selection or analysis purposes at all, correct? If so, please clarify in text (I see that the figure says they are for illustrative purposes).

Indeed, these categories were used to ensure that we sampled from a wide range of everyday visual experience when collecting stimuli; however, they do not enter into the analysis. We have clarified this point in the Methods on page 17.

Pg 8 - related to the above it is not clear in the main text how r_{cv} is calculated across the two sets of videos. From reading the methods, I believe that r_{cv} is calculated separately for each half of the data and then only voxels that are found to have $r_{cv} > 0$ in both analyses are selected. Please clarify.

You are correct: the model is used to predict the data from video sets 1 and 2 separately, resulting in two calculations of r_{cv} (1 per set). We also conducted separate clustering analyses for each video set. We have updated the Results and Methods sections to clarify this point (pg. 7, 18)

Pg 6 - For both action and effector PCA analyses you describe the first PC in detail. Were the other PCs similarly interpretable? I see the schematics in Figure S1, but it is a bit hard to interpret these alone.

In this manuscript we decided to leave most of the interpretation of these principle components to the reader, as there is no single correct interpretation of these results, and none of our claims hinge on the interpretation of the principle components.

Figure 4- what is r_{cv} ? Is it the same thing as the model prediction correlation presented in the top left? Please clarify in figure legend and text.

Yes, r_{cv} is the same as the prediction correlation. We have clarified our use of this term in the Figure 4 legend and the Results (pg. 7).

Pg 9 - Network 1 appears to be tuned only to mouths (not other face features) based on the coloring in Figure 4. Is this correct and what are the implications of this?

Network 1 is tuned to both mouths and the head in general, but not the eyes, nose, or ears. This is most likely because the socially-relevant actions in our stimulus set (e.g., smiling, laughing, talking) saliently engage the mouth and cheeks, but not the nose or eyes. It is possible that this exact profile might change if we included social actions that manipulate the nose or eyes more, such as winking or wrinkling one's nose in disgust at another person; however, we cannot confidently interpret the implications of these details without running another study with new stimuli.

Reviewer #2 (Remarks to the Author):

This study by Tarhan & Konkle looked at the functional organization of visual actions across cerebral cortex in humans. They use voxel-wise encoding models to predict brain activity in response to held-out videos of humans performing everyday actions, and show that these models are strikingly similar across two sets of videos. In addition, they cluster the data in order to interpret the networks of areas that represent this information.

Overall, this study is quite thorough. The methods show clearly that the results are replicable. The results are interesting. However, the specific methods are suboptimal in several respects. Given that voxel-wise modeling is relatively rarely used in fMRI it is important that the methods used in any voxel-wise modeling report should be beyond reproach. I think it is likely that these suggestions will not change the overall results of the paper, but they will make the paper stronger.

Major Issues:

This is a vision science paper, and it has been standard practice in the vision community for many years to present all of the data from each individual subject. Showing data from individual subjects avoids confounds due to cortical normalization and it increases confidence in the results. This paper claims to be using the same voxel-wise encoding model pipelines as Mitchell et al. (2008) and Huth et al. (2012), so showing data from individual subjects should not be a problem. Given the amount of data collected on each subject, it should be possible to show the effect in each subject, and if not, the authors should report how many of the individual subjects do show the effect.

We have added single-subject results for the body-part-action-target model's performance and the data-driven clustering analysis to the supplement (pg. 30, Figure S4). We have also included single-subject reliability maps (pg. 30, Figure S2) and ordered the results in Figure S2 and Figure S4 according to the extent of their reliable coverage.

There are some major problems with how PCA is used in this study. Typically, during voxel-wise modeling PCA is used as a dimensionality reduction technique during data interpretation, in situations with a very large number of variables/features. However, in this paper PCA is used as part of the model fitting process. However, it is not necessary to use PCA to fit the body part and target feature spaces used here because they have only 20 and 5 features, respectively. All fitting and interpretation involving those features should be done in the native space, which were explicitly labeled by participants.

While PCA is often used to reduce the risk of over-fitting, we used it to address the reality that some of the hand-labeled features are highly correlated with each other (e.g., each of the fingers). Using regularized regression on a "native" feature space would ensure that these correlated features were not used to fine-tune the model too much, and effectively would apply an appropriate weight to one of the finger features while setting the rest to zero. However, which finger was given a non-zero weight would be essentially arbitrary. When we clustered voxels by their profile of regression weights, we might then find that some clusters were tuned to a single finger – and in fact, that is exactly what we found. It was thus difficult to interpret these tuning profiles when the model was fit to the native feature space. Applying PCA to the feature space

groups these colinear features together, thus avoiding the situation where correlated features are assigned very different weights. When we interpreted the clusters' tunings, we then translated these weights back into the native feature space – we calculated the cluster's tuning for each finger based on how strongly it loaded onto each principle component. Therefore, using PCA rendered our analysis more stable and still allowed us to interpret our results within the native feature space.

To investigate how the model prediction is affected by this step, we compared our results to a model that was fit to the native feature space. We found very little difference in prediction performance (average difference in cross-validated $r = 0.02$, $sd = 0.11$ for set 1; average = 0.01, $sd = 0.11$ for set 2). We have added these findings and clarified our motivation for using PCA in the Supplement (pg. 30).

Because the Gist model has many parameters PCA might offer a way to reduce computation time during model fitting. However, if it is feasible to fit those models without PCA then this would be preferable. If PCA is needed later in order to interpret the feature tuning of voxels fit well by that model, then PCA should be run on the fit model weights, not the feature space.

Finally, if the authors find it impossible to fit the Gist model without using PCA, then they should justify the number of features used, as they did with the other feature spaces (e.g. 95% variance explained). In the current version of the manuscript, it is unclear whether the Gist model is fitting worse than the higher-level action feature model in some brain areas merely because the PC features do not capture enough of the variance in the specific features that are encoded in those areas. Note that this suggestion may reduce the number of voxels that prefer the higher-level model, but it is impossible to predict by how much these results might change.

Here, PCA is actually a crucial step in defining the Gist features, as outlined in Oliva & Torralba (2001). These features are meant to capture information about the *global* layout and spatial frequency of a scene. The global nature of the gist features is only possible after the original features – individual gabors applied at different locations throughout the video frame – are PC'd. Without this step, the model would be based on highly local information, rather than features that describe the scene's structure or "shape," such as the presence of a horizon or the predominance of vertical orientations. Similarly, the choice to include the first 20 PC's (which together account for nearly 100% of the variance in the videos) is based on the procedure from Oliva & Torralba. We have updated the Methods (pg. 18) to include these details.

The authors claim that they apply PCA to the stimulus features during fitting in order to remove correlations between variables. However, this is completely unnecessary because they use ridge regression for model fitting. Ridge regression deals with the problem of correlated variables directly. In fact, because ridge regression uses regularization and PCA does not, in the presence of noise the ridge procedure is likely to do a better job than PCA. For this reason the PCA step should be removed from the modeling procedure. This will also make it easier to interpret the fit model weights in terms of the feature spaces of interest.

As we described above, ridge regression reduces the risk that the model will use correlated variables to over-fit the data; however, it does not solve the problem of how to interpret the resulting weights for correlated predictors.

I expect that these recommended changes in the use of PCA will not impact the conclusions of they study, but there is a chance that they will increase model performance for all models. In any case, these analyses should be corrected in order to conform to best statistical practices.

Please see our response to your first comment about the use of PCA above.

When using regularized regression to fit models to noisy data, the regularization parameter often has the largest effect on the outcome. Therefore, it is critically important to ensure that the data set used for regularization and model fitting is completely separate from the data set used for evaluation of the fit models. However, as far as I can tell the current manuscript does not enforce this critical separation. Instead, the regularization coefficient through k-fold cross validation on the training data (which is fine), but then the model is evaluated based on the same data that was used to select the regularization coefficient (for one set of stimuli). The authors of the paper already have two sets of data, so it would be incredibly straightforward to fit the model on set 1 of the stimuli and then to use the second set of stimuli to evaluate the model.

We have changed our modeling procedure following your prescription, and have updated the Methods (pg. 19) and Results (pg. 7) accordingly. In addition, all figures based on modeling results have been updated (Fig. 4, Fig.5, Fig. 6, Fig. 7, S3, S4, S5, S7, and S9).

In revision the authors should also provide a more convincing test of model performance. Separating fit and test data sets in this way has proven to be a much more robust test of a model than the leave one out cross validation (LOOCV) procedure that the authors used in the current submission. Again, the correct way to deal with this issue is to fit the data on set 1 of the stimuli and then predict responses to set 2 of the stimuli. Thus, separating sets 1 and 2 of the stimuli for model fitting and model testing would solve both of these problems simultaneously.

We have changed our modeling procedure following your prescription. The Methods (pg. 19), Results (pg. 7), and all relevant results figures (Fig. 4, Fig.5, Fig. 6, Fig. 7, S3, S4, S5, and S9) have been updated to reflect this change. Note that our overall interpretations were unchanged, as the performance of our body-part-action-target model was similar with both procedures (mean improvement in \$r = 0.08\$, \$sd = 0.15\$ for video set 1; mean improvement = \$0.06\$, \$sd = 0.14\$ for video set 2).

Finally, I should note that the threshold used for "well-fit" voxels appears to be rather arbitrary. It is currently defined as those with $rCV > 0$, but statistical significance should be determined empirically. The most straightforward way to do this would be to get the p-values associated with each voxel's rCV value. These values should then be FDR-corrected, and the voxels with values remaining above threshold should be further analyzed. This change should clean up the cluster analyses, and it is likely that fewer voxels will be colored gray in Figure 4 after this threshold is revised.

We have altered our analysis to select voxels following your prescription. The Methods (pg. 19), Results (pg. 9) and relevant figures (Fig. 4, Fig. 5, S3, S4, S5, and S7) have been updated accordingly.

Minor Issues:

All figures with inflated brains should show both left and right hemispheres, as well as medial views. This is important not only for completeness, but also because the paper makes reference to results in medial areas (such as retrosplenial cortex), which are not actually visible in any of the main figures of the paper.

We have added right-hemisphere and medial views to Figures 2, 4, 5, 6, 7b, S2, S4, S5, and S9.

It is unclear where the model weights for clustering come from, particularly because models are fit 60 times during LOOCV. I assume that these are the average model weights across all of these folds, but that is not stated explicitly in the paper. This information should be included in the methods.

We have added this to the Methods (pg. 19) to explain that the model weights were found using the full dataset in our updated analysis (which no longer uses LOOCV).

From the methods, it seems that the body part features do not differentiate the left and the right sides of the body. If this is true then it should be added to the methods for clarity.

We have clarified this point in the Methods (pg. 18).

Reviewers' Comments:

Reviewer #1:

Remarks to the Author:

The authors have addressed most of my concerns. And I think the paper is much improved. I still have two remaining concerns that I believe should be addressed before publication:

First, I am still somewhat unclear on the feature selection procedure. My understanding is that the authors:

- 1) Calculate the split-half reliability of each voxel (either within or across video sets)
- 2) For each threshold -1:1 select the voxels with a reliability above that threshold
- 3) Compute the item-pattern reliability of the above-threshold voxels. (I don't believe that item pattern reliability is defined anywhere in the text. Is it the correlation between a given voxel's response to all videos in set 1 and set 2? Please add this definition)
- 4) Somehow select identify the point where a more stringent threshold no longer improves reliability. This is the step I am particularly confused about. How is this threshold selected? Is it by finding an elbow/taking the derivative of the curve? Another method?

Second, your variance partitioning results are interesting. I am not familiar with variance suppression referenced. However, it seems like this analysis differs in some key ways from that in the paper. 1) It is ROI-based (this is not really an issue if it will not be included in the paper). 2) More importantly, body parts and action targets are separated whereas these models are combined in Figure 7B. It seems like you see a different pattern for body parts than action targets in certain ROIs (e.g., LOTC visible bodies > involved bodies, but involved actions > visible actions). Do you think this difference is meaningful? What do the results look like if you test on combined (body parts + targets) visible vs. involved models?

Reviewer #2:

Remarks to the Author:

The authors have clearly put a lot of work into their revision, and it is clear that the manuscript has improved considerably since the first submission. In particular, the comparison of models fit on video sets 1 and 2 is quite compelling, as are some of the individual subject results. However, there are still a few points that need to be addressed. In the following document the authors' comments are quoted, and my replies are given in normal text.

Major Issues:

"While PCA is often used to reduce the risk of over-fitting, we used it to address the reality that some of the hand-labeled features are highly correlated with each other (e.g., each of the fingers). Using regularized regression on a "native" feature space would ensure that these correlated features were not used to fine-tune the model too much, and effectively would apply an appropriate weight to one of the finger features while setting the rest to zero. However, which finger was given a non-zero weight would be essentially arbitrary. When we clustered voxels by their profile of regression weights, we might then find that some clusters were tuned to a single finger – and in fact, that is exactly what we found. It was thus difficult to interpret these tuning profiles when the model was fit to the native feature space. Applying PCA to the feature space groups these colinear features together, thus avoiding the situation where correlated features are assigned very different weights."

This description of the problem has led me to believe that the authors are instead using "lasso" and not "ridge" for regularization, as this is exactly the expected outcome of lasso. If there are two perfectly correlated features in a model, lasso will select one of those features arbitrarily to be zero, but ridge will instead split the total beta value evenly across the two features. This is a well-known phenomenon, and often is a reason why ridge is chosen over lasso in these circumstances. A brief look at the code shows that they had options to run lasso regression, so perhaps this is what was actually presented in the paper. There are also mentions of Elastic Net, which will result in a mix between the weights obtained from lasso and ridge. Elastic Net is also not ideal for this study because the arbitrary splitting of weights will still occur, though to a lesser degree than with lasso. Among these options, ridge is the best suited to deal with the highly correlated features which are used in this study. In addition, I note that the methods section of this manuscript says that ridge was used, not lasso or Elastic Net. The authors should verify that this is accurate.

It may help the authors to review the differences between lasso, ridge, and Elastic Net when dealing with perfectly correlated regressors. There may be a simple bug in their code that can be detected by creating a toy dataset with just two identical regressors, and then using these three regularization methods to obtain quite different sets of weights.

The bottom line is that this outcome should not occur when ridge is used. If this problem persists after the authors have verified that they are using ridge and not lasso, there is almost certainly a problem with the regression code. Once this is resolved, PCA should be removed from this location in the analysis pipeline, for the

reasons stated in my previous review.

"We have added single-subject results for the body-part-action-target model's performance and the data-driven clustering analysis to the supplement (pg. 30, Figure S4). We have also included single-subject reliability maps (pg. 30, Figure S2) and ordered the results in Figure S2 and Figure S4 according to the extent of their reliable coverage."

The model performance maps provided in Figure S4 are compelling because they show that in individual subjects, the same general areas of cortex are well-predicted by the model. However, the same does not seem to be true of the clustering results across individuals, and this is rather concerning. Even when looking at the subjects with the most reliable coverage (Sub 1 & Sub 2), the majority of parietal and temporal cortex seem to have different tuning across these two subjects. In fact, I am struggling to find any two subjects that have similar clustering maps. If the results presented here do not appear consistent across subjects, that is something that needs to be thoroughly discussed and justified in the manuscript. Otherwise, it is probable that the group-averaged results are heavily influenced by a few individuals, and in this case it would be likely that the authors are drawing their conclusions from biased data that is not actually shared across their subjects.

Related to this, it seems that somewhat different colors for each cluster are used on the cortical surfaces for different subjects. For example, the yellow seen in the right hemisphere of Sub 5 does not seem to be present on any other subject, and the purples used across subjects are also a variety of different hues. There are a number of inconsistencies apparent, so the authors should check their code closely and then regenerate these maps to ensure that they are using the same colormap for every subject.

It is possible that these two problems are related, and that when the authors take care to plot the data on all subjects consistently, the differences in tuning may be somewhat resolved. But that is unclear at this time and will need to be re-evaluated after revision.

Minor Issues:

Upon another review of the methods section, it seems that the range of regularization parameters tested has not been included. Inclusion of that information would be helpful for future re-analysis or replication.

Reviewer #1 (Remarks to the Author):

The authors have addressed most of my concerns. And I think the paper is much improved. I still have two remaining concerns that I believe should be addressed before publication:

First, I am still somewhat unclear on the feature selection procedure. My understanding is that the authors:

- 1) Calculate the split-half reliability of each voxel (either within or across video sets)
- 2) For each threshold -1:1 select the voxels with a reliability above that threshold
- 3) Compute the item-pattern reliability of the above-threshold voxels. (I don't believe that item pattern reliability is defined anywhere in the text. Is it the correlation between a given voxel's response to all videos in set 1 and set 2? Please add this definition)

Item pattern reliability is the correlation for an individual action (e.g., running), calculated across the selected voxels in odd and even runs. We have clarified this in the text (Figure 2 caption; pg. 19).

- 4) Somehow select identify the point where a more stringent threshold no longer improves reliability. This is the step I am particularly confused about. How is this threshold selected? Is it by finding an elbow/taking the derivative of the curve? Another method?

You're correct, this threshold is selected by finding the plateau point of the reliability curve—this is the threshold at which continuing to restrict the voxels no longer adds to item-level multivoxel pattern reliability. We used this plateau point to select a reliability threshold of $r > 0.30$ by eye, though you are correct that this could also be done by taking a derivative. This method was recently published in *NeuroImage* (<https://doi.org/10.1016/j.neuroimage.2019.116350>), and also has an accompanying code package with example data available on the Open Science Framework (<https://osf.io/m9ykh/>).

Second, your variance partitioning results are interesting. I am not familiar with variance suppression referenced. However, it seems like this analysis differs in some key ways from that in the paper. 1) It is ROI-based (this is not really an issue if it will not be included in the paper).

This is true. In our last response, we included a graph of one of the commonality analyses we had done for due-diligence, to show that we had in fact tried a form of commonality analysis and ended up with uninterpretable negative variance estimates. And, while we did this particular analysis at the level of ROIs, rather than voxels, we included this result in our response to illustrate this challenge, which indicated to us that the commonality analysis was not suited to our data. As you noted above, we don't plan on including this in these uninterpretable results in the paper, so we agree that the ROI-based nature of this commonality analysis is not an issue.

- 2). More importantly, body parts and action targets are separated whereas these models are combined in Figure 7B. It seems like you see a different pattern for body parts than action targets in certain ROIs (e.g., LOTC visible bodies > involved bodies, but involved actions > visible actions). Do you think this difference is meaningful? What do the results look like if you test on combined (body parts + targets) visible vs. involved models?

This comment follows up on the graph we included in our first response (which is not included in the revised manuscript). To answer your question, we do not think the difference you are referring to above is meaningful. This is because, one of the feature spaces – body part involvement – accounted for a *negative* amount of variance. As variance partitioning methods for fMRI data like this are relatively new, we're still working through understanding what could be happening here (after conversations with Mark Lescroart and Mick Bonner, who have published papers using this method, we think there might be a contribution from variance suppression, but we are not confident about that). Luckily, we think that the main take-homes of the manuscript are not contingent on the results of a commonality analysis.

In our revised manuscript, we highlight that Figure 7b simply shows which model predicts the data best. We also explicitly note that this analysis does not indicate how much variance is shared between these models and that attempts at commonality (or “variance partitioning”) analyses yielded unstable results (pg. 14).

Reviewer #2 (Remarks to the Author):

The authors have clearly put a lot of work into their revision, and it is clear that the manuscript has improved considerably since the first submission. In particular, the comparison of models fit on video sets 1 and 2 is quite compelling, as are some of the individual subject results.

Thank you!

However, there are still a few points that need to be addressed. In the following document the authors' comments are quoted, and my replies are given in normal text.

Major Issues:

"While PCA is often used to reduce the risk of over-fitting, we used it to address the reality that some of the hand-labeled features are highly correlated with each other (e.g., each of the fingers). Using regularized regression on a "native" feature space would ensure that these correlated features were not used to fine-tune the model too much, and effectively would apply an appropriate weight to one of the finger features while setting the rest to zero. However, which finger was given a non-zero weight would be essentially arbitrary. When we clustered voxels by their profile of regression weights, we might then find that some clusters were tuned to a single finger – and in fact, that is exactly what we found. It was thus difficult to interpret these tuning profiles when the model was fit to the native feature space. Applying PCA to the feature space groups these colinear features together, thus avoiding the situation where correlated features are assigned very different weights."

This description of the problem has led me to believe that the authors are instead using "lasso" and not "ridge" for regularization, as this is exactly the expected outcome of lasso. If there are two perfectly correlated features in a model, lasso will select one of those features arbitrarily to be zero, but ridge will instead split the total beta value evenly across the two features. This is a well-known phenomenon, and often is a reason why ridge is chosen over lasso in these circumstances. A brief look at the code shows that they had options to run lasso regression, so perhaps this is what was actually presented in the paper. There are also mentions of Elastic Net, which will result in a mix between the weights obtained from lasso and ridge. Elastic Net is also not ideal for this study because the arbitrary splitting of weights will still occur, though to a lesser degree than with lasso. Among these options, ridge is the best suited to deal with the highly correlated features which are used in this study. In addition, I note that the methods section of this manuscript says that ridge was used, not lasso or Elastic Net. The authors should verify that this is accurate.

It may help the authors to review the differences between lasso, ridge, and Elastic Net when dealing with perfectly correlated regressors. There may be a simple bug in their code that can be detected by creating a toy dataset with just two identical regressors, and then using these three regularization methods to obtain quite different sets of weights.

The bottom line is that this outcome should not occur when ridge is used. If this problem persists after the authors have verified that they are using ridge and not lasso, there is almost

certainly a problem with the regression code. Once this is resolved, PCA should be removed from this location in the analysis pipeline, for the reasons stated in my previous review.

Thank you for your comments – we now see that our original response was unclear and resulted in a misunderstanding. The reviewer is actually critiquing a hypothetical which we had introduced in our response as a possible illustrative case—this was not a pattern found in our data. As you noted, we did indeed use ridge regression in our analyses.

To return to your original concern, you wrote, “However, it is not necessary to use PCA to fit the body part and target feature spaces used here because they have only 20 and 5 features, respectively.” We agree! However, while not necessary, it is also not incorrect. We think you’d agree with us that removing this step is unlikely to yield any meaningful changes in this paper’s results or interpretation.

However, in response to these thoughtful comments, in our revised manuscript we have clarified that PCA was not a necessary step in our analysis pipeline, for anyone interested in adapting our methods (Supplemental Methods section, pg. 30).

"We have added single-subject results for the body-part-action-target model’s performance and the data-driven clustering analysis to the supplement (pg. 30, Figure S4). We have also included single-subject reliability maps (pg. 30, Figure S2) and ordered the results in Figure S2 and Figure S4 according to the extent of their reliable coverage."

The model performance maps provided in Figure S4 are compelling because they show that in individual subjects, the same general areas of cortex are well-predicted by the model. However, the same does not seem to be true of the clustering results across individuals, and this is rather concerning. Even when looking at the subjects with the most reliable coverage (Sub 1 & Sub 2), the majority of parietal and temporal cortex seem to have different tuning across these two subjects. In fact, I am struggling to find any two subjects that have similar clustering maps. If the results presented here do not appear consistent across subjects, that is something that needs to be thoroughly discussed and justified in the manuscript. Otherwise, it is probable that the group-averaged results are heavily influenced by a few individuals, and in this case it would be likely that the authors are drawing their conclusions from biased data that is not actually shared across their subjects.

Related to this, it seems that somewhat different colors for each cluster are used on the cortical surfaces for different subjects. For example, the yellow seen in the right hemisphere of Sub 5 does not seem to be present on any other subject, and the purples used across subjects are also a variety of different hues. There are a number of inconsistencies apparent, so the authors should check their code closely and then regenerate these maps to ensure that they are using the same colormap for every subject.

It is possible that these two problems are related, and that when the authors take care to plot the data on all subjects consistently, the differences in tuning may be somewhat resolved. But that is unclear at this time and will need to be re-evaluated after revision.

Thank you for drawing our attention to this, as this figure was indeed hard to interpret. In the revised manuscript, we have made two changes: (1) We remade all the single subject figures in Supplemental Figure 4 so that the clusters' color similarity is now interpretable across participants, and included the corresponding cluster center profiles (see also updates to the Supplemental Methods, pg. 30). (2) We also now comment on the single subject patterns of data in the main manuscript (pg. 10), indicating both patterns of robustness and of variation, and we encourage readers to interpret these single-subject results for themselves.

Minor Issues:

Upon another review of the methods section, it seems that the range of regularization parameters tested has not been included. Inclusion of that information would be helpful for future re-analysis or replication.

We have added this information to the Methods (pg. 19).

Reviewers' Comments:

Reviewer #1:

Remarks to the Author:

The authors have addressed all of my concerns.

Reviewer #2:

Remarks to the Author:

The authors have addressed the vast majority of my concerns with the manuscript, and I believe it has improved greatly during the review process. At this point, I have just one final request before I believe it is ready for publication.

As seen in all previous rounds of review, the authors and I disagree about the statistical practice of performing PCA on the features prior to regression. We agree that the inclusion of that step is quite unlikely to change the overall conclusions of this paper, and the authors admit that this step was not necessary. However, they have still left this step in their analysis despite the concerns I outlined in my initial review (reproduced below):

"The authors claim that they apply PCA to the stimulus features during fitting in order to remove correlations between variables. However, this is completely unnecessary because they use ridge regression for model fitting. Ridge regression deals with the problem of correlated variables directly. In fact, because ridge regression uses regularization and PCA does not, in the presence of noise, ridge is likely to do a better job than PCA. For this reason the PCA step should be removed from the modeling procedure. This will also make it easier to interpret the fit model weights in terms of the feature spaces of interest. I expect that these recommended changes in the use of PCA will not impact the conclusions of this study, but there is a chance that they will increase model performance for all models. In any case, these analyses should be corrected in order to conform to best statistical practices."

I understand that changing the analysis pipeline at this stage may be difficult for the authors. Therefore, I will be willing to accept the manuscript if the authors include a note in the methods explaining that the use of PCA is not only unnecessary, but that it also likely hurt model prediction accuracy to some degree. This can easily be addressed by adding a sentence or two on page 30 of the manuscript, where the authors already inserted the other information about PCA in the previous revision.

Few people in our field have attempted voxelwise modeling, and it is important to give potential readers a sense of which processing steps are likely optimal versus suboptimal when performing these kinds of analyses.

Reviewer #1 (Remarks to the Author):

The authors have addressed all of my concerns.

Thank you for your helpful comments!

Reviewer #2 (Remarks to the Author):

The authors have addressed the vast majority of my concerns with the manuscript, and I believe it has improved greatly during the review process. At this point, I have just one final request before I believe it is ready for publication.

As seen in all previous rounds of review, the authors and I disagree about the statistical practice of performing PCA on the features prior to regression. We agree that the inclusion of that step is quite unlikely to change the overall conclusions of this paper, and the authors admit that this step was not necessary. However, they have still left this step in their analysis despite the concerns I outlined in my initial review (reproduced below):

"The authors claim that they apply PCA to the stimulus features during fitting in order to remove correlations between variables. However, this is completely unnecessary because they use ridge regression for model fitting. Ridge regression deals with the problem of correlated variables directly. In fact, because ridge regression uses regularization and PCA does not, in the presence of noise, ridge is likely to do a better job than PCA. For this reason the PCA step should be removed from the modeling procedure. This will also make it easier to interpret the fit model weights in terms of the feature spaces of interest. I expect that these recommended changes in the use of PCA will not impact the conclusions of this study, but there is a chance that they will increase model performance for all models. In any case, these analyses should be corrected in order to conform to best statistical practices."

I understand that changing the analysis pipeline at this stage may be difficult for the authors. Therefore, I will be willing to accept the manuscript if the authors include a note in the methods explaining that the use of PCA is not only unnecessary, but that it also likely hurt model prediction accuracy to some degree. This can easily be addressed by adding a sentence or two on page 30 of the manuscript, where the authors already inserted the other information about PCA in the previous revision.

Few people in our field have attempted voxelwise modeling, and it is important to give potential readers a sense of which processing steps are likely optimal versus suboptimal when performing these kinds of analyses.

We agree that it is important to help readers to understand how to implement a voxelwise modeling analysis. To that end, we have added to the Supplementary Methods (pg. 3) to explain that PCA is not necessary when using ridge regularization and can even negatively impact the model's prediction accuracy to some degree.